# The Small GTPase Ran Increases Sensitivity of Ovarian Cancer Cells to Oncolytic Vesicular Stomatitis Virus

**DOI:** 10.3390/ph17121662

**Published:** 2024-12-10

**Authors:** Karen Geoffroy, Mélissa Viens, Emma Mary Kalin, Zied Boudhraa, Dominic Guy Roy, Jian Hui Wu, Diane Provencher, Anne-Marie Mes-Masson, Marie-Claude Bourgeois-Daigneault

**Affiliations:** 1Cancer Axis, Centre de Recherche du Centre Hospitalier de l’Université de Montréal and Institut du Cancer de Montréal, Montreal, QC H2X 0A9, Canadamelissa.viens@umontreal.ca (M.V.); emma.kalin@umontreal.ca (E.M.K.); dominicguyroy@gmail.com (D.G.R.); diane.provencher.med@ssss.gouv.qc.ca (D.P.);; 2Department of Microbiology, Infectious Diseases and Immunology, Faculty of Medicine, University of Montreal, Montreal, QC H3T 1J4, Canada; 3Department of Medicine, Faculty of Medicine, University of Montreal, Montreal, QC H3T 1J4, Canada; 4Gerald Bronfman Department of Oncology, Faculty of Medicine, McGill University, Montreal, QC H3A 0G4, Canada; jian.h.wu@mcgill.ca; 5Division of Gynecologic Oncology, Faculty of Medicine, University of Montreal, Montreal, QC H3T 1J4, Canada

**Keywords:** oncolytic viruses, VSV, ovarian cancer, RanGTP

## Abstract

**Background/Objectives:** Ovarian cancer is the deadliest gynecologic cancer, and with the majority of patients dying within the first five years of diagnosis, new therapeutic options are required. The small guanosine triphosphatase (GTPase) Ras-related nuclear protein (Ran) has been reported to be highly expressed in high-grade serous ovarian cancers (HGSOCs) and associated with poor outcomes. Blocking Ran function or preventing its expression were shown to be promising treatment strategies, however, there are currently no small molecule inhibitors available to specifically inhibit Ran function. Interestingly, a previous study suggested that the Vesicular stomatitis virus (VSV) could inhibit Ran activity. Given that VSV is an oncolytic virus (OV) and, therefore, has anti-cancer activity, we reasoned that oncolytic VSV (oVSV) might be particularly effective against ovarian cancer via Ran inhibition. **Methods**: We evaluated the sensitivity of patient-derived ovarian cancer cell lines to oVSV, as well as the impact of oVSV on Ran and vice versa, using overexpression systems, small interfering RNAs (siRNAs), and drug inhibition. **Results**: In this study, we evaluated the interplay between oVSV and Ran and found that, although oVSV does not consistently block Ran, increased Ran activation allows for better oVSV replication and tumor cell killing. **Conclusions**: Our study reveals a positive impact of Ran on oVSV sensitivity. Given the high expression of Ran in HGSOCs, which are particularly aggressive ovarian cancers, our data suggest that oVSV could be effective against the deadliest form of the disease.

## 1. Introduction

Epithelial ovarian cancer (EOC) is an aggressive disease for which the late-stage forms show a poor survival rate of only 29% [1]. The current standard of care treatment includes an aggressive debulking surgery, either preceded or followed by a combination of taxane- and platinum-based chemotherapy [2]. Unfortunately, more than 70% of advanced ovarian cancers become refractory to standard treatments, underscoring the need for the development of new therapies [3].

Due to specific cellular alterations that characterize malignant transformation, OVs preferentially infect and replicate in tumor cells [4,5,6]. Various OVs are being tested worldwide, and one, a Herpes simplex virus, is currently approved for metastatic melanoma treatment in North America, Europe, and Australia [7]. Another promising OV candidate is VSV [8]. More specifically, VSVΔ51, an oncolytic mutant of VSV engineered to harbor a deletion of the methionine in position 51 of its matrix protein to abolish its capacity to counteract cellular antiviral pathways that are often defective in cancer cells [9]. This modification allows for the selective replication of VSVΔ51 in tumor cells and improves the safety profile of the treatment [10]. Although oVSV is effective in ovarian cancer [11,12,13,14], cell-intrinsic factors can affect treatment efficacy. Indeed, while many cancers evolved defects in antiviral response pathways [15,16], which confers a benefit for immune evasion, OVs and others developed altered cellular metabolism [17] to support rapid proliferation, which favors viral replication; not all cancers have such characteristics and the factors governing OV sensitivity versus resistance are not fully understood.

We have previously demonstrated that the small GTPase Ran is overexpressed in HGSOCs [18], the most common form of the disease [19]. Of particular interest, elevated Ran expression has been associated with tumor transformation and progression [20,21] and correlates with poor outcomes in HGSOCs [18]. Pre-clinical studies by us and others have established the benefits of blocking Ran in both ovarian [22,23] and breast cancer as a novel treatment strategy [24]. There is currently no available drug on the market with known Ran inhibitory activity. However, VSV has been suggested to have this effect [25]. Given that VSV also has direct anti-cancer activity as an OV [10,26,27], we tested if it could block Ran activity in ovarian cancer, which could confer an ideal therapy against the disease. Using a collection of EOC cell lines of various subtypes, we unexpectedly found that oVSV does not consistently inhibit Ran activation. Interestingly, we also observed that Ran could promote oVSV replication and virus-mediated cancer killing. To our knowledge, this is the first report of the effect of Ran on virus infection. Our data suggest that Ran expression could be used as a biomarker to select which ovarian cancer patients should undergo OV therapy. As such, cancers like HGSOCs in which Ran is often overexpressed [18], could be good candidates.

## 2. Results

### 2.1. Most Human Ovarian Cancer Cells Are Sensitive to oVSV

First, we wanted to characterize our cell lines for their viral sensitivities and Ran expression. We previously published that most human EOC cell lines were sensitive to oVSV infection [14]. Here, we first sought to determine their sensitivities to virus-mediated cancer killing in a panel of 11 ovarian cancer cell lines of all subtypes (see Appendix A for details). When infecting the cells with oVSV-YFP at a multiplicity of infection (MOI) of 0.1, we observed a heterogeneous response to infection, with some cell lines being highly infectable, while some others showed no fluorescence 24 h post-infection (Figure 1A), a result that is in line with our previous findings [14]. As expected, we also observed various sensitivities to virus killing, with most cells showing high viral sensitivities with drops in viability of >50% at 48 h post-infection (Figure 1B). Notably, 3 cell lines (OV3331, TOV2414, and TOV3392D) were particularly resistant to virus killing with viabilities greater than 40% 72 h post-infection.

We next performed a Western blot analysis to measure basal Ran expression in the different cell lines. Comparing expression levels across cell lines can be challenging due to the varying constitutive expression of common proteins used as loading controls. Therefore, we loaded equivalent protein concentrations for all cell lines, ensured similar phenol red signals for all conditions (Appendix A), and used 2 different proteins (GAPDH and vinculin) as loading controls. Although the bands corresponding to GAPDH and vinculin are not perfectly consistent across conditions, most cell lines show similar signals (Figure 1C). Interestingly, the Ran signal was lower in 4 cell lines compared to the others. Of these, 3 were the most resistant cell lines to virus-mediated cancer killing, as shown in Figure 1B, and the other was among the less sensitive ones. Notably, all virus-sensitive cells had higher Ran expression, therefore suggesting a link between Ran expression and intrinsic sensitivity to oVSV.

### 2.2. oVSV Does Not Modulate Ran Activation

In a previous study, Her et al. [25] reported an impact of the matrix protein of VSV on the nuclear export of RNAs and suggested that this effect was the result of Ran inhibition. However, the impact of VSV on Ran activity was never directly explored. Here, we sought to determine if oVSV could inhibit Ran activation, which was assessed by measuring GTP-bound Ran (active form of Ran) [28] using a pull-down assay. To do so, EOC cell lines were infected with oVSV, and Ran and RanGTP were measured by Western blot (Figure 2). Given the heterogeneous sensitivities of the different EOC cell lines to oVSV, different MOIs were used to ensure that all cells were infected to comparable levels. MOIs were selected based on the viral sensitivity of each cell line, as determined in Figure 1. High MOIs were used to ensure the infection of all cells and samples were collected 8 h post-infection to avoid virus-mediated cell killing (Appendix A).

Surprisingly, we found that Ran activation was not consistently modulated by oVSV. Indeed, infection had either no impact on Ran activation (TOV3133G, TOV2414, and TOV1946 (Figure 2A)), decreased (TOV2835EP, TOV3392D, and TOV112D (Figure 2B)), or increased (TOV3041G, OV3331, TOV21G, OV1946, and OV2085 (Figure 2C)) Ran activation. Notably, we found no association between the ovarian cancer subtype or previous treatments and the viral sensitivity and the effect of infection on Ran (Appendix A). Importantly, the basal expression of Ran was not affected by infection, except in the TOV112D cell line, in which it was slightly increased. Our data, therefore, show that oVSV does not consistently modulate Ran activation.

### 2.3. Ran Inhibition Impairs VSV Replication

Given our finding that oVSV-resistant cells express lower levels of Ran (Figure 1C), we next wanted to measure the impact of Ran on VSV. We selected EOC cell lines with various sensitivities to oVSV (as seen in Figure 1). To determine the impact of Ran inhibition on virus infection, we first silenced Ran expression by siRNA transfection. Ran knock-downs (KD) were confirmed by Western blot (Appendix A). The cells were infected 40 h post-siRNA transfection. Once again, different MOIs were selected to account for the heterogeneous viral sensitivities of the different cell lines. Given that we are analyzing virus infection and production 24 h post-infection, we selected lower MOIs to ensure that oVSV-sensitive cell lines were not completely killed by the virus at the time of sample collection. Interestingly, when measuring the fluorescent signal post-oVSV-YFP infection, we found cells to be less infected with Ran KD (Figure 3A).

In line with this, quantification of virus outputs from culture supernatants confirmed that fewer viral particles were produced with Ran KD compared to control conditions for all the cell lines tested (Figure 3B). Given that Ran inhibition eventually leads to cancer cell death [22], we wanted to determine if the decreased viral replication we observed could be explained by poor cell viability at the time of infection. Cell viability was therefore quantified at various time points post-siRNA transfection for each cell line. While cells were left uninfected for these experiments, the time points corresponding to infection and sample collection in Figure 3 are indicated on the graphs (Appendix A), and we found no significant differences in cell viability at these time points. We also found Ran KD to decrease cell viability at later time points, a finding that is in line with our previous study [22].

To validate that Ran inhibition impairs oVSV, we used a Ran-blocking drug. M36 has been shown to bind to Ran and to decrease its GTP-activation, as well as to specifically induce cell death in aneuploid cancer cells (as detailed in our published patent WO2019046931A1). M36 is currently in pre-clinical testing for ovarian cancer treatment. To determine if M36-mediated Ran inhibition has a similar effect as Ran KD, 3 cell lines of various viral sensitivities (TOV112D, TOV1946, and OV3331) were pre-treated with the drug for 16 h and then infected with oVSV. As expected, we observed decreased fluorescent signals (Figure 4A), as well as decreased viral outputs (Figure 4B) for all cell lines with M36 treatment.

As was done for Ran KD, we measured the viability of cells after M36 treatment at the times of infection and sample collection and found that all cell lines were viable at our experimental time points (Appendix A). We also confirmed that, as expected, M36 treatment kills cancer cells at later time points. We repeated our experiment to measure the impact of earlier drug treatment by simultaneously inhibiting Ran and infecting the cells, and similar results were obtained in these conditions (Appendix A). Taken together, our results show that inhibiting Ran dampens oVSV infection and replication.

### 2.4. Ran Inhibition Impairs oVSV-Mediated Tumor Cell Killing

Given that oVSV has cancer-killing abilities, we quantified cell viability upon infection in the context of Ran inhibition. Consistent with impaired virus replication, cellular viability was higher in infected cells with Ran KD (Figure 5A). We also observed a significant decrease in cancer killing for 2 out of 3 cell lines tested when M36 was administered before or simultaneously with oVSV infection (Figure 5B and Appendix A). Taken together, our data show that decreased Ran expression impairs tumor cell killing by oVSV.

### 2.5. Ran Enhances oVSV

Given that Ran blockade impairs oVSV and that low Ran expression is found in virus-resistant cell lines (Figure 1C), we next wanted to determine if increased Ran activation could enhance virus replication, as well as its oncolytic capacities. To measure this, we used 2 cell lines, TOV1946 and OV3331, which are sensitive and resistant to oVSV infection, respectively. Cells were transfected with constitutive active (CA) or wild type (WT) Ran constructs. The CA Ran is a mutant for which the glutamine in position 69 was substituted for a leucine, which inhibits its intrinsic hydrolytic activity and, therefore, allows for Ran to remain in an active, GTP-bound state [29]. Our results show that, as expected, CA Ran allowed for increased virus production and enhanced cancer cell killing compared to control conditions (Figure 6A and Figure 6B, respectively).

## 3. Discussion

In this study, we evaluated the interplay between Ran and oVSV. Our results disagree with the previously suggested inhibition of Ran activity by VSV by Her et al. [18]. Notably, the authors of this previous study showed that the matrix protein of VSV inhibited the nuclear export of RNAs but did not directly measure Ran activation. Subsequent studies by other groups demonstrated that the matrix protein interacts with the nucleoporin 98 (Nup98), a phenylalanine-glycine-rich nucleoprotein involved in the nucleocytoplasmic transport of different types of RNAs, rather than directly binding Ran [30]. Interestingly, other viruses have been shown to interact with Nup98. As such, the severe acute respiratory syndrome coronavirus 2 interacts with Nup98, which results in the blockade of the interferon pathway [31]. Additionally, the human immunodeficiency virus requires Nup98 during its early phases of replication [32,33,34,35]. Notably, we measured Ran activation by using a Ran pull-down assay that allows for the quantification of GTP-bound Ran. This assay has been widely used in several studies [36,37,38] but measures active Ran rather than its activity and there are no assays commercially available to measure Ran activity. Nevertheless, it would be interesting to assess the impact of oVSV on Ran activity by measuring Ran-mediated RNA transport as others did in previous studies [39,40], but in the context of viral infection by oVSV.

An exciting finding of our study is that Ran allows for enhanced oVSV replication. Given that OVs are currently being explored as treatments for different types of cancer [41,42,43], the increased viral replication we observe with high Ran expression could have therapeutic implications. However, given that we show that some EOC cell lines are refractory to oVSV, the virus should not be used as a stand-alone treatment for all ovarian cancer patients. However, given that HGSOCs express high levels of Ran [12], which correlates with the worst prognoses, the most aggressive forms of the disease could, therefore, respond particularly well to VSV therapy. Notably, we have previously shown that combining oVSV with Janus kinase (JAK) inhibitors was effective in the most virus-resistant EOC cell lines. JAK inhibitors are drugs that prevent signaling by antiviral interferons [44] and have also been shown by others to enhance OVs in different types of cancer [45,46,47,48]. We, therefore, believe that oVSV could be effective as a therapy for EOC either in the context of high Ran expression and/or in combination with JAK inhibitors. One limitation of our study is that all experiments were conducted in EOC cell lines. Whether our findings translate to other cancer types, as well as using clinical samples, remains to be tested. Additionally, future work will investigate the molecular mechanisms governing the effect of Ran on viral sensitivity.

In this study, we show that Ran inhibition counteracts oVSV and, therefore, could reduce OV treatment efficacy. M36 is an anti-cancer drug that is not currently used to treat patients but is expected to move to clinical testing. Importantly, our findings indicate that M36 and oVSV therapies are not compatible and should not be used concurrently to treat cancer patients.

Beyond ovarian cancer, other neoplasms for which Ran overexpression is observed could benefit from our findings. As such, adrenal gland, bladder, thyroid, brain, esophagus, stomach, colon, ovary, uterus, pancreas, skin, kidney, liver, lung, breast, testis, prostate, and cervix cancers that classically overexpress Ran [49] could benefit from oVSV therapy. Therefore, our results show that overexpression of Ran is not only a biomarker associated with bad prognosis but could also be used to select patients to treat with OVs. Patient samples from these different cancers and at various stages should be tested for correlation between Ran expression and oVSV sensitivity prior to clinical implementation. Ex vivo infection in microfluidic systems could be used to conduct these experiments, as we have in a previous study [14].

## 4. Materials and Methods

### 4.1. Cell Lines and Culture

All ovarian cancer cell lines used in this study are a part of the OvCAN collection and are available to the Canadian research community for pre-clinical studies upon request (https://ovariancanada.org/wp-content/uploads/2023/12/1-Dec-2023_OvCAN-Collection-v4.pdf, accessed on 1 December 2024). All cell lines used in this study have been previously described [14,50,51,52,53,54], and characteristics can be found in Appendix A. Primary ovarian cancer cell lines were derived from aggressive EOC patient samples and include serous (TOV1946, OV1946, TOV3041G, TOV3133G, OV2085, TOV2835EP), clear cell (TOV3392D, TOV21G), endometrioid (TOV112D), mucinous (TOV2414), and unspecified (OV3331) adenocarcinomas. Cells were cultured in OSE medium (Wisent Inc., Saint-Bruno, QC, Canada) supplemented with 10% FBS (Wisent), 2.5 µg/mL amphotericin B (Wisent), and 50 µg/mL gentamicin (Gibco, Billings, MT, USA) and kept at 37 °C with 5% CO_2_. Kidney epithelial Vero cells (ATCC, Manassas, VA, USA) were cultured in DMEM medium (Gibco) supplemented with 10% FBS (Wisent) and kept at 37 °C with 5% CO_2_.

### 4.2. Virus Imaging and Quantification

The OV used in this study is a VSV variant of the Indiana strain. More specifically, we used an oncolytic mutant (VSVΔ51, oVSV) deleted for the methionine 51 of its matrix protein [7] and encoding either the yellow fluorescent protein (YFP) or the red fluorescent protein (RFP), both of which have already been described [55,56]. For fluorescence analysis, live high-resolution pictures were acquired using a ZOE fluorescent cell imager (Biorad, Hercules, CA, USA), and high-throughput screens were performed using an EnSight multimode plate reader (Perkin Elmer, Waltham, MA, USA) and its associated software (Kaleido 3.0).

For virus quantification, titers from culture supernatants were measured by plaque assays as described previously [57]. Briefly, infectious samples were serially diluted and transferred onto confluent monolayers of Vero cells, and an agarose overlay was applied. Plaques were counted 24 h later.

### 4.3. Viability Assays

Coomassie blue stains were used as a measure of cell viability, as described previously [58]. This technique stains the cells in the wells, and given that the cells detach upon oVSV killing, a stronger Coomassie blue signal corresponds to higher cell viability. We favor this readout over other techniques, such as MTS and lactate dehydrogenase (LDH) assays that measure metabolic activity (MTS [59] and LDH assay [60]), which can be modulated by virus infection without necessarily killing the cells [61]. Briefly, cells were fixed using a solution of acetic acid/methanol (1:3) for 30 min and stained with a solution of 0.1% Coomassie blue (in acetic acid/methanol 1:3) for 30 min. Plates were then washed with tap water and allowed to dry at room temperature. The next day, a solution of 1% sodium dodecyl sulfate (Bioshop, Burlington, ON, Canada) was added to the wells for 1 h, and the signal was quantified at 595 nm using an EnSight multimode plate reader (Perkin Elmer) and its associated software (Kaleido 3.0).

For some experiments, as mentioned in the figure legends, cell viability was measured using trypan blue. To do so, cell suspensions were diluted 1:2 in trypan blue (Sigma-Aldrich, Saint-Louis, MO, USA), and viability was measured using a TC20 automated cell counter (Biorad).

### 4.4. Ran Activation Assays

Ran activation was measured using the Ran activation assay kit from Cell Biolabs (San Diego, CA, USA), which quantifies RanGTP, the active form of the protein. Samples were generated by infecting monolayers of tumor cells for 8 h at the indicated multiplicities of infection (MOIs) before harvesting and processing according to the manufacturer’s protocol.

### 4.5. Western Blot

Cell pellets were lysed using RIPA buffer (25 mM Tris-HCl pH 7.6, 150 mM NaCl, 5 mM EDTA, 1% Triton X-100, 1% sodium deoxycholate, 0.1% SDS) supplemented with complete protease inhibitors (Sigma-Aldrich), sonicated for 1 s/10 µL using a Misonix XL-2000 Series ultrasonic liquid processor (American laboratory trading, East Lyme, CT, USA), and incubated on ice for 1 h. Lysates were then centrifuged at 12,000× *g* for 10 min, and cleared supernatants were used for analysis. Protein concentrations were determined using a DC protein assay kit (5000111, Biorad) according to the manufacturer’s protocol. Ladder (1610375, Biorad) and samples were migrated on 12% acrylamide gels and transferred onto 0.45 µm nitrocellulose membranes (1620115, Biorad). Membranes were blocked in milk (5%) for 1 h at room temperature and probed with rabbit anti-human Ran (ab53775, Abcam, Cambridge, UK), rabbit anti-human GAPDH (14C10, Cell signaling technology, Danvers, MA, USA), rabbit anti-vinculin (E1E9V, Cell signaling technology), or rabbit anti-VSV (house-made [27]) overnight at 4 °C. The next day, membranes were washed in TBS (0.02 M Tris base, 0.15 M NaCl)-Tween 0.05% and incubated for 1 h with a goat anti-rabbit-HRP IgG (7074S, Cell signaling technology). All antibodies were diluted in TBS-5% milk. Membranes were once again washed in TBS-Tween 0.05%, and signals were revealed using the Clarity Max Western ECL Substrate (1705062, Biorad) and a Chemidoc imaging system (Biorad).

### 4.6. Transfections

Monolayers of tumor cells were transfected and infected at the indicated MOIs 48 h post-transfection. siRNAs were obtained from Dharmacon (Lafayette, CO, USA) (human Ran siRNAs: 5′-AGAAGAAUCUUCAGUACUA-3′, 5′-GUGAAUUUGAGAAGAAGUA-3′, 5′-CCUAUUAAGUUCAAUGUAU-3′, and 5′-ACAGGAAAGUGAAGGCGAA-3′ and control siRNAs: 5′-UAGCGACUAAACACAUCAA-3′, 5′-UAAGGCUAUGAAGAGAUAC-3′, 5′-AUGUAUUGGCCUGUAUUAG-3′, and 5′-AUGAACGUGAAUUGCUCAA-3′). siRNAs were transfected using RNAiMax lipofectamine (Thermofisher Scientific, Waltham, MA, USA) as per the manufacturer’s protocols. GFP-tagged Ran wild type (WT) and constitutively active (CA, RanQ69L) constructs have been described previously [62]. Plasmids were amplified using the plasmid plus maxi kit (12963, Qiagen, Hilden, Germany) as per the manufacturer’s protocol. Plasmids were transfected using lipofectamine 2000 (Thermofisher Scientific) according to the manufacturer’s protocol.

### 4.7. M36 Drug and Treatment

M36, a novel Ran-targeting drug (patent WO2019046931A1) [63], was used at the final concentration of 40 µM. Unless specified otherwise, cells were treated 16 h prior to infection, and samples were collected 24 h post-infection.

### 4.8. Statistical Analyses

Statistical analyses were performed using GraphPad Prism 9.0. Unpaired multiple *t*-tests were performed to compare the different groups. Results were considered significant when *p*-values were inferior to 0.5.

## 5. Conclusions

In this study, we show the sensitivity of EOC cell lines to oVSV infection and killing, especially when cell lines highly express Ran, and demonstrate for the first time that Ran enhances oVSV. These findings could be harnessed to improve VSV therapy for ovarian cancer and lead to improvements in patient responses to OVs.

## Figures and Tables

**Figure 1 pharmaceuticals-17-01662-f001:**
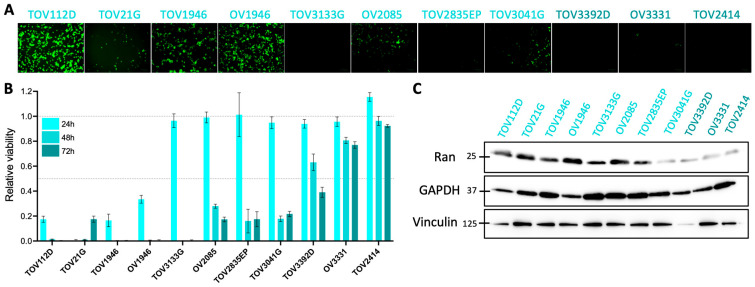
oVSV infects and kills human ovarian cancer cells. (**A**) Representative fluorescence pictures of TOV112D, TOV21G, TOV1946, OV1946, TOV3133G, OV2085, TOV2835EP, TOV3041G, TOV3392D, OV3331, and TOV2414 cells infected with oVSV-YFP at an MOI of 0.1 for 24 h. (**B**) Relative cell viabilities 24, 48, and 72 h post-infection with oVSV at an MOI of 0.1 (n = 6). Dotted lines highlight 100% and 50% viabilities. (**C**) Ran expression of ovarian cancer cells as measured by Western blot. GAPDH and vinculin are protein-loading controls.

**Figure 2 pharmaceuticals-17-01662-f002:**
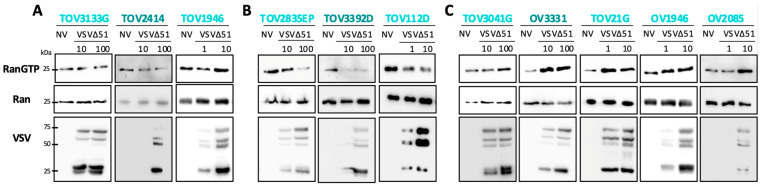
oVSV infection does not consistently inhibit Ran. Ran activation assay measuring RanGTP (activated Ran) and total Ran (used here as a loading control) by Western blot upon Ran pull-down. A polyclonal antibody against VSV, recognizing multiple viral proteins, was also used to detect infection. Cells were either left untreated or infected with oVSV-YFP at the indicated MOIs for 8 h. MOIs used for each cell line were selected based on the viral sensitivities determined in Figure 1. (**A**–**C**) show different effects of infection on RanGTP expression. NV = no virus.

**Figure 3 pharmaceuticals-17-01662-f003:**
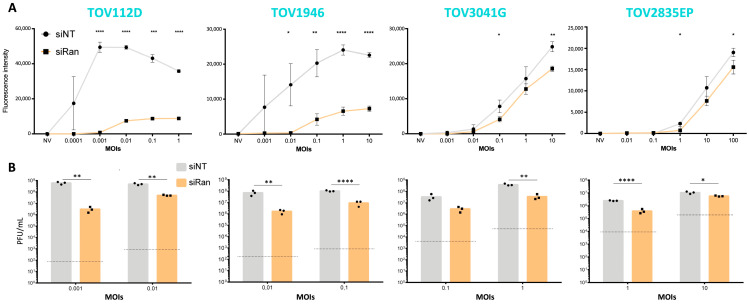
Ran KD decreases VSV replication. (**A**) Fluorescence intensities of TOV112D, TOV1946, TOV3041G, and TOV2835EP cells transfected with non-targeting (NT) or Ran-targeting siRNAs and infected 16–24 h later with oVSV-YFP at the indicated MOIs (n = 3). (**B**) Virus outputs (plaque forming units (PFUs)) were measured 24 h post-infection (n = 3). The dotted lines represent viral inputs. Unpaired multiple *t*-test: *: *p* ≤ 0.05; **: *p* ≤ 0.01; ***: *p* ≤ 0.001; ****: *p* ≤ 0.0001.

**Figure 4 pharmaceuticals-17-01662-f004:**
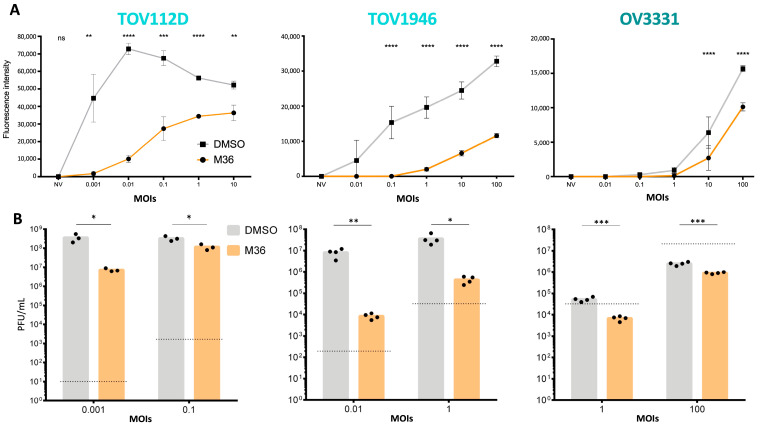
Ran inhibition impairs VSV replication. (**A**) Fluorescent intensities of TOV112D, TOV1946, and OV3331 cells treated or not with M36 (40 µM) for 16 h prior to infection with oVSV-YFP at the indicated MOIs (n ≥ 3). (**B**) Virus outputs were quantified 24 h post-infection (n ≥ 3). The dotted lines represent viral inputs. Statistical analyses by unpaired multiple *t*-test: ns: *p* > 0.05; *: *p* ≤ 0.05; **: *p* ≤ 0.01; ***: *p* ≤ 0.001; ****: *p* ≤ 0.0001.

**Figure 5 pharmaceuticals-17-01662-f005:**
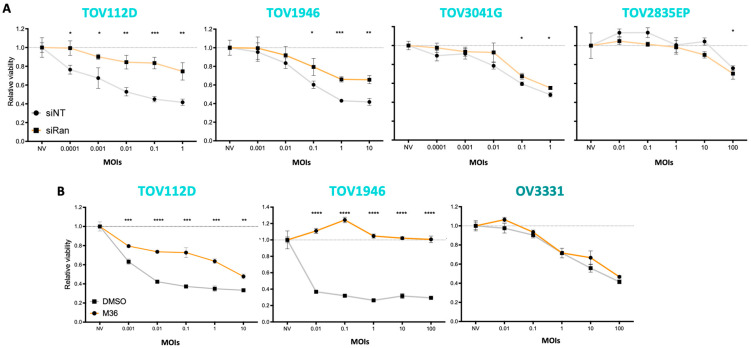
Ran KD and inhibition impair oVSV-mediated cancer killing. Relative viability of (**A**) TOV112D, TOV1946, TOV3041G, and TOV2835EP cells transfected with control non-targeting (NT) or Ran-targeting siRNAs and infected with oVSV-YFP at various MOIs for 24 h (n = 3) or (**B**) TOV112D, TOV1946, and OV3331 cells pre-treated with M36 (40 µM) for 16 h and infected with oVSV-YFP at various MOIs for 24 h (n = 3). The dotted lines represent viability in non-infected control conditions. Unpaired multiple *t*-test: *: *p* ≤ 0.05; **: *p* ≤ 0.01; ***: *p* ≤ 0.001; ****: *p* ≤ 0.0001.

**Figure 6 pharmaceuticals-17-01662-f006:**
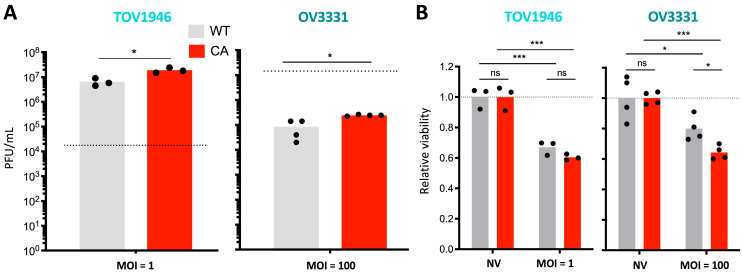
Ran activation enhances oVSV. (**A**) Virus outputs and (**B**) cell viability of TOV1946 and OV3331 cells transfected with constitutive active (CA) or wild type (WT) Ran constructs and infected with oVSV-RFP 24 h later (n ≥ 3). Samples were collected 24 h post-infection. Dotted lines represent viral inputs (**A**) or viability in non-infected control conditions in (**B**). Unpaired multiple *t*-test: ns: *p* > 0.05; *: *p* ≤ 0.05, ***: *p* ≤ 0.001.

## Data Availability

Cell lines are a part of the OvCAN collection and are available to the Canadian research community for pre-clinical studies upon request.

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
