# Peer review of "The Small GTPase Ran Increases Sensitivity of Ovarian Cancer Cells to Oncolytic Vesicular Stomatitis Virus"

_pharmaceuticals, 2024, doi:10.3390/ph17121662_

Round 1
Reviewer 1 Report
Comments and Suggestions for Authors
I have reviewed the manuscript titled "The small GTPase Ran increases the viral sensitivity of ovarian cancer cells to oncolytic Vesicular stomatitis virus" with the manuscript ID: pharmaceuticals-3327500. The authors aimed to explore the potential of VSV in targeting Ran activity in ovarian cancer cell lines as a therapeutic strategy. The study revealed that oVSV did not consistently inhibit Ran activation; rather, it was observed that Ran could enhance VSV replication and the virus-induced cancer cell death in ovarian cancer cell lines.
1. On page 3, line 49, it is recommended that the authors provide a brief explanation of how these "cell intrinsic factors" impact the effectiveness of cancer treatment.
2. Abbreviations should be defined in the text where they appear for the first time.
3. Why did the authors use a higher MOI for viral infection in some cell lines compared to others? Is it acceptable to use different MOIs when comparing different groups?
4. It is recommended to provide more detailed explanations in the caption of Figure 2.
5. In Fig. 6B, it is recommended that the authors compare each of the CA Ran or WT Ran groups with their respective non-infected control cells.
6. In the Results section on page 14, why did the authors only mention the overexpressed CA Ran as the factor that increased virus production and enhanced cancer cell killing, while WT Ran also significantly improved these factors?
7. In the end of Dissection section, pages 249-251, to accurately assess the potential of a protein like Ran as a cancer biomarker, its expression must be tested in clinical samples obtained from patients at various stages of cancer. Therefore, the related sentence should be changed.
8. In Materials and Methods section, Viability assays page 7, why didn't the authors use more common and accurate experiments, such as MTS assay, BrdU test, LDH assay, etc., for viability assay?
Overall, in my opinion, the manuscript is acceptable for publishing with minor revision based on the comments mentioned in this review report and those outlined in the manuscript uploaded on the journal's website.
Author Response
Reviewer 1
“I have reviewed the manuscript titled "The small GTPase Ran increases the viral sensitivity of ovarian cancer cells to oncolytic Vesicular stomatitis virus" with the manuscript ID: pharmaceuticals-3327500. The authors aimed to explore the potential of VSV in targeting Ran activity in ovarian cancer cell lines as a therapeutic strategy. The study revealed that oVSV did not consistently inhibit Ran activation; rather, it was observed that Ran could enhance VSV replication and the virus-induced cancer cell death in ovarian cancer cell lines.
- On page 3, line 49, it is recommended that the authors provide a brief explanation of how these "cell intrinsic factors" impact the effectiveness of cancer treatment.”
We would like to thank the reviewer for his thorough assessment of our paper and insightful comments. We have now expanded on cell intrinsic factors that can impact effectiveness of OV-therapy. We discuss antiviral response defects, as well as cellular metabolism and mention that many factors remain unknown (lines 50-55).
“2. Abbreviations should be defined in the text where they appear for the first time.”
We would like to thank the reviewer for pointing this out. We have now revised the manuscript to ensure that all abbreviations are defined when they first appear in the text.
“3. Why did the authors use a higher MOI for viral infection in some cell lines compared to others? Is it acceptable to use different MOIs when comparing different groups?”
We now realize that we did not justify the use of different MOIs in the original version of our manuscript. Indeed, while Figure 1 characterizes the viral sensitivities of all EOC cell lines used in the paper at the same MOI across cell lines, we do use different MOIs for different cell lines starting in Figure 2. We did so to ensure conditions with comparable levels of infection. The data obtained in Figure 1, as well as from our previous study using these cell lines were used to determine the MOIs we used in subsequent experiments.
Given that we are not comparing one cell line to another, but rather conditions using the same cell line, we believe that comparable infection was more important in this context. For instance, if we used the same MOIs with all cell lines, sensitive cells would have been completely killed by the virus as soon as 18h post-infection. Also, if we used low MOIs, this would not have allowed for infection in the most resistant cell lines.
This information is now found in the revised manuscript (lines 116-118 and 147-151).
“4. It is recommended to provide more detailed explanations in the caption of Figure 2.”
We have now modified to legend of Figure 2 to include the type of assay used, as well as what each antibody measures and a justification of the chosen MOIs (lines 123-129 and 147-151).
“5. In Fig. 6B, it is recommended that the authors compare each of the CA Ran or WT Ran groups with their respective non-infected control cells.”
We would like to thank the reviewer for this relevant comment that we have now addressed in the text. We now compare both conditions with their non-infected controls and have also added the stats to the graphs.
“6. In the Results section on page 14, why did the authors only mention the overexpressed CA Ran as the factor that increased virus production and enhanced cancer cell killing, while WT Ran also significantly improved these factors?”
The graphs from Figure 6 only compare cells that were transfected with WT Ran vs CA Ran. The untransfected condition was not included because there was no difference with WT Ran and we believe that WT Ran is a better control. The dashed lines indicate inputs and there is no comparison with untransfected cells. Therefore, CA Ran increased oVSV production, not WT Ran. This is why we did not comment on WT Ran in the text.
“7. In the end of Dissection section, pages 249-251, to accurately assess the potential of a protein like Ran as a cancer biomarker, its expression must be tested in clinical samples obtained from patients at various stages of cancer. Therefore, the related sentence should be changed.”
We would like to thank the reviewer for this comment. We have now addressed this in the discussion. “Patient samples from these different cancers and at various stages should be tested for correlation between Ran expression and oVSV sensitivity prior to clinical implementation.” (lines 289-290)
“8. In Materials and Methods section, Viability assays page 7, why didn't the authors use more common and accurate experiments, such as MTS assay, BrdU test, LDH assay, etc., for viability assay?”
We understand that this is an important question and other groups prefer to use different assays. However, we believe that using Coomassie blue as a readout of cell viability is more accurate in the context of VSV infection because most other readouts use metabolic assays (MTS, LDH), which could be modulated by virus infection even without oncolysis and therefore would not reflect cancer killing. Adherent cells detach upon oVSV infection and killing, which decreases the Coomassie blue signal. We have used this technique reliably in many studies. This is now explained in the text (lines 325-329).
“Overall, in my opinion, the manuscript is acceptable for publishing with minor revision based on the comments mentioned in this review report and those outlined in the manuscript uploaded on the journal's website.”

Reviewer 2 Report
Comments and Suggestions for Authors
The authors investigated the relationship between the GTPase Ran, which is overexpressed in HGSOCs (high-grade serous ovarian cancers), and the oncolytic VSV (oVSV), which has anti-cancer activity. The major findings are: 1) oVSV infects and kills human epithelial ovarian cancer (EOC) cells with different efficiency – three EOC cell lines showed resistance to oVSV killing (no or less oVSV killing); four EOC cell lines that expressed low levels of Ran were associated with less oVSV killing; 2) oVSV infection does not consistently inhibit Ran levels; 3) Ran knock-down (KD) decreased oVSV infection and replication; 4) Ran inhibition by M36 impairs oVSV replication. 5) Ran KD and inhibition by M36 impaired oVSV-mediated cancer killing. 6) Ran activation enhances oVSV replication and killing. The authors concluded that Ran increases the sensitivity to oVSV killing; as HGSOCs have elevated Ran levels, oVSV could be effective against HGSOCs. Overall, the data supported the findings and conclusions, with the limitation that all analyses were performed in EOC cell lines, which should be mentioned as a limitation of the study in Discussion.
A few points to consider:
1. Title: “viral sensitivity” may be confusing; “sensitivity” to oncolytic VSV may read better.
2. M36 was introduced as an anti-cancer drug candidate, which works by inhibiting Ran. As oVSV kills cancer cells with high levels of Ran, it may be worth mentioning that M36 and oVSV should not be used together for cancer treatment.
3. Figure 6, the enhancement by Ran activation was only minimal – this should be clearly stated.
4. It may be useful to comment that, given the varying MOIs and sensitivities to oVSV in EOC cell lines, practically speaking, what is the chance for oVSV to be developed for cancer treatment?
Author Response
Reviewer 2
"The authors investigated the relationship between the GTPase Ran, which is overexpressed in HGSOCs (high-grade serous ovarian cancers), and the oncolytic VSV (oVSV), which has anti-cancer activity. The major findings are: 1) oVSV infects and kills human epithelial ovarian cancer (EOC) cells with different efficiency – three EOC cell lines showed resistance to oVSV killing (no or less oVSV killing); four EOC cell lines that expressed low levels of Ran were associated with less oVSV killing; 2) oVSV infection does not consistently inhibit Ran levels; 3) Ran knock-down (KD) decreased oVSV infection and replication; 4) Ran inhibition by M36 impairs oVSV replication. 5) Ran KD and inhibition by M36 impaired oVSV-mediated cancer killing. 6) Ran activation enhances oVSV replication and killing. The authors concluded that Ran increases the sensitivity to oVSV killing; as HGSOCs have elevated Ran levels, oVSV could be effective against HGSOCs. Overall, the data supported the findings and conclusions, with the limitation that all analyses were performed in EOC cell lines, which should be mentioned as a limitation of the study in Discussion.”
We would like to thank the reviewer for this comment. Indeed, we only used EOC cell lines in our experiments, which is a limitation of our study, this is now mentioned in the discussion of our revised manuscript (lines 274-276).
“ A few points to consider:
- Title: “viral sensitivity” may be confusing; “sensitivity” to oncolytic VSV may read better.”
We have now changed the title to: The small GTPase Ran increases sensitivity of ovarian cancer cells to oncolytic Vesicular stomatitis virus.
“2. M36 was introduced as an anti-cancer drug candidate, which works by inhibiting Ran. As oVSV kills cancer cells with high levels of Ran, it may be worth mentioning that M36 and oVSV should not be used together for cancer treatment.”
Our results indeed indicate that M36 should not be given at the same time as oVSV for cancer treatment. We have now modified the discussion of the manuscript to elaborate on this aspect (lines 279-282) and would like to thank the reviewer for the suggestion.
“3. Figure 6, the enhancement by Ran activation was only minimal – this should be clearly stated.”
The reviewer is right, the enhancement is rather modest. We have modified the text to better reflect this (lines 235-236).
“4. It may be useful to comment that, given the varying MOIs and sensitivities to oVSV in EOC cell lines, practically speaking, what is the chance for oVSV to be developed for cancer treatment?
This is an important point. Our revised manuscript now discusses this issue. We now state that oVSV should not be used to treat all EOC cancer patients and reiterate that Ran expression could be used for patient selection. We also mention a treatment combination that could work better than oVSV alone based on our previous study on EOC (lines 264-266 and 268-273).
